# SinGAN-Seg: Synthetic training data generation for medical image segmentation

**Vajira Thambawita**[1,2]ᵒ\*, **Pegah Salehi**[1]ᵒ, **Sajad Amouei Sheshkal**[1]ᵒ, **Steven A. Hicks**[1], **Hugo L. Hammer**[1,2], **Sravanthi Parasa**[6], **Thomas de Lange**[3,4,5], **Pål Halvorsen**[1,2], **Michael A. Riegler**[1]ᵒ

**1** SimulaMet, Oslo, Norway, **2** Oslo Metropolitan University, Oslo, Norway, **3** Medical Department, Sahlgrenska University Hospital-Möndal, Gothenburg, Sweden, **4** Department of Molecular and Clinical Medicine, Sahlgrenska Academy, University of Gothenburg, Gothenburg, Sweden, **5** Augere Medical, Oslo, Norway, **6** Department of Gastroenterology, Swedish Medical Group, Seattle, WA, United States of America

ᵒ These authors contributed equally to this work.

\* vajira@simula.no

**Data Availability Statement:** The synthetic dataset is freely available at https://osf.io/xrgz8/as an example synthetic dataset generated using the SinGAN-Seg pipeline. The pre-trained deep learning

## Abstract

Analyzing medical data to find abnormalities is a time-consuming and costly task, particularly for rare abnormalities, requiring tremendous efforts from medical experts. Therefore, artificial intelligence has become a popular tool for the automatic processing of medical data, acting as a supportive tool for doctors. However, the machine learning models used to build these tools are highly dependent on the data used to train them. Large amounts of data can be difficult to obtain in medicine due to privacy reasons, expensive and time-consuming annotations, and a general lack of data samples for infrequent lesions. In this study, we present a novel synthetic data generation pipeline, called *SinGAN-Seg*, to produce synthetic medical images with corresponding masks using a single training image. Our method is different from the traditional generative adversarial networks (GANs) because our model needs only a single image and the corresponding ground truth to train. We also show that the synthetic data generation pipeline can be used to produce alternative artificial segmentation datasets with corresponding ground truth masks when real datasets are not allowed to share. The pipeline is evaluated using qualitative and quantitative comparisons between real data and synthetic data to show that the style transfer technique used in our pipeline significantly improves the quality of the generated data and our method is better than other state-of-the-art GANs to prepare synthetic images when the size of training datasets are limited. By training UNet++ using both real data and the synthetic data generated from the SinGAN-Seg pipeline, we show that the models trained on synthetic data have very close performances to those trained on real data when both datasets have a considerable amount of training data. In contrast, we show that synthetic data generated from the SinGAN-Seg pipeline improves the performance of segmentation models when training datasets do not have a considerable amount of data. All experiments were performed using an open dataset and the code is publicly available on GitHub.

models are available at https://github.com/vlbthambawita/singan-seg-polyp.

**Funding:** The author(s) received no specific funding for this work.

**Competing interests:** I have read the journal's policy and the authors of this manuscript have the following competing interests: Sravanthi Parasa: Consultant Covidien LP; Medical advisory board-Fujifilms. Thomas de Lange: Share holder and employed (20%) Augere medical. Pål Halvorson: Board member of Augere Medical.

## Introduction

Artificial intelligence (AI) has become a popular tool in medicine and has been vastly discussed in recent decades to improve the performance of clinicians [1–4]. According to the statistics discussed by Jiang et al. [1], artificial neural networks (ANNs) [5], and support vector machines (SVMs) [6] are the most popular machine learning (ML) algorithms used with medical data. Among the various applications of AI in medicine, medical image analysis [7–9] has become a popular research area applying such ML methods. In this respect, ML models learn from data; thus, the amount and quality of medical data has a direct influence on the success of ML-based applications. While the SVM algorithms are popular for regression [10, 11] and classification [12] tasks, ANNs or deep neural networks (DNNs) are are used widely for all types of ML tasks; regression, classification, detection, and segmentation.

Segmentation models make more advanced predictions than regression, classification, and detection as it performs pixel-wise classification of the input images. Therefore, medical image segmentation is a popular application of AI in medicine for image analysis [13–15]. Image segmentation can help find the exact regions that delineate a specific lesion, which could be polyps in gastrointestinal (GI) images, skin cancer in images of the skin, brain tumors in magnetic resonance imaging (MRI), and many more. However, the success of segmentation models is highly dependent on the size and quality of the data used to train them, which is normally annotated by experts like medical doctors.

In this regard, we identified three main reasons why there are mostly small public datasets in the medical domain compared to other domains. The first reason is privacy concerns attached with medical data containing potentially sensitive patient information. The second is the costly and time-consuming medical data annotation processes that the medical domain experts must perform. Finally, the third is the rarity of some abnormalities.

Privacy concerns can vary across countries and regions according to the data protection regulations present in the specific areas. For example, Norway should follow the rules given by the Norwegian data protection authority (NDPA) [16] and enforce the personal data act [17], in addition to following the general data protection regulation 31 (GDPR) [18] guidelines being the same for all European countries. While there is no central level privacy protection guideline in the US like GDPR in Europe, US rules and regulations are enforced through other US privacy laws, such as Health Insurance Portability and Accountability Act (HIPAA) [19] and the California Consumer Privacy Act (CCPA) [20]. In Asian counties, they follow their own sets of rules, such as Japan's Act on Protection of Personal Information [21], the South Korean Personal Information Protection Commission [22] and the Personal Data Protection Bill in India [23]. Additionally, if research is performed with such privacy restrictions, the papers published are often theoretical methods only. According to the analyzed medical image segmentation studies in [24], 30% have used private datasets. As a result, the studies are not reproducible. Furthermore, universities and research institutes that use medical data for teaching purposes use the same medical datasets for years, which affects the quality of education.

In addition to the privacy concerns, the costly and time-consuming medical data labeling and annotation process [25] is an obstacle to producing large datasets for AI algorithms. Compared to other already time-consuming medical data labeling processes, pixel-wise data annotation is far more demanding in terms of time. If the data annotations by experts are not possible, experts should do at least a review process to make the annotations correct before using them in AI algorithms. The importance of having accurate annotations from experts for medical data is, for example, discussed by [26] using a mandible segmentation dataset of computed tomography (CT) images. In this regard, researching a way to produce synthetic segmentation datasets (synthetic images and the corresponding accurate ground truth masks) to

extend the training datasets is important to overcome the timely and costly medical data annotation process.

Synthetic data is a possible solution to overcome the privacy issues related to medical image data and reduce the cost and time needed for annotations especially in combination with differential privacy [27–29] which is important for medical datasets that can include patient identifying data such as images from faces of stroke patients. Synthetic data generated by generative adversarial networks (GANs) is used in almost all domains that use ML, including the medical domain [30–34], agriculture [35, 36], and robotics [37–39]. Among these, some studies [33–36, 40–42] generate only synthetic data, while other studies [30, 31, 33] generate synthetic data and the corresponding ground truth. [30] generated synthetic glioblastoma multiforme (GBM) in 2D magnetic resonance images with a segmentation mask. For this method, they manually placed artificially generated GBM in real images using MeVisLab (https://www.mevislab.de), calling it semi-supervised data generation. However, placing the generated synthetic GBM is a time-consuming task for generating a big synthetic dataset using this approach. Moreover, the background image is still real while the segmented GBM is synthetic. The real sections of synthetic images may raise privacy concerns.

Another study [31], used the conditional GAN to generate synthetic polyp data conditioned on real edge filtering images and a randomly generated mask. This approach can generate diverse synthetic data, but a large dataset is required to train this generative model because it uses the deep convolution GAN (DCGAN) [43] which is data-hungry to train. Moreover, extracted edge-filtering images and random polyp masks should be merged to generate the corresponding synthetic data. Therefore, input data preparation is a time-consuming process.

Instead of generating synthetic polyps using edge filtering, we previously used real non-polyp images and random polyp masks to convert non-polyp images into polyp images using a GAN-based image inpainting model [33]. In the latter two cases, a random mask should be provided to generate the corresponding synthetic polyp image. Moreover, in the image inpainting GAN for polyps, generated synthetic polyps are unrealistic as a result of locating them randomly. Therefore, a thorough post-screening process is required.

For the above methods, a considerable amount of manually annotated data is needed, for which the time-consuming process of manual data annotation is required. Therefore, we present an alternative synthetic data generation process, which can be used to extend small datasets with an unlimited number of synthetic data samples and corresponding ground truth masks without any manual process.

As a case study in this paper, we use polyp segmentation, which is a popular segmentation task that uses ML techniques to detect and segment polyps in images/videos collected from GI examinations, such as colonoscopy or capsule endoscopy. Early identification of polyps in GI tract is critical to prevent colorectal cancers [44]. Therefore, many ML models have been investigated to automatically segment polyps in GI tract videos recorded from both colonoscopies [45–47] or capsule endoscopy examinations [48–50], with the aim of decreasing the miss rates and reducing the inter- and intra-observer variations.

Most polyp segmentation models are based on convolutional neural networks (CNNs) and are trained using publicly available polyp segmentation datasets [51–55]. However, these datasets have a limited number of images with corresponding expert annotated segmentation masks. For examples, the CVC-VideoClinicDB [52] dataset has 11, 954 images from 10 polyp videos and 10 non-polyp videos with segmentation masks, the PICCOLO dataset [55] has 3, 433 manually segmented images (2, 131 white-light images and 1, 302 narrow-band images), and the Hyper-Kvasir [51] dataset has only 1, 000 images with the corresponding segmentation masks, but also contains 100, 000 unlabeled images. In this regard, researching an alternative way, which is applicable with small datasets, to generate synthetic data to tackle the various

challenges that we previously discussed, is the main objective of this study. The contributions of this paper are as follows:

- This study introduces the SinGAN-Seg pipeline to generate synthetic medical images and their corresponding segmentation masks using a single image as training data. This method is different from traditional GAN methods, which often need large training datasets. A modified version of the state-of-the-art SinGAN architecture with a fine-tuning step using a style-transfer method is used. We use polyp segmentation as a case study, but the SinGAN-Seg can be applied for all types of segmentation tasks.

- We compare our method with different other generative methods and benchmark them specifically for GAN performance when only little amount of data is available for the training process.

- We present the largest synthetic polyp dataset with the corresponding masks and make it publicly available online at https://osf.io/xrgz8/. Moreover, we have published our generators as a python package at Python package index (PyPI) (https://pypi.org/project/singan-seg-polyp/) to generate an unlimited number of polyps and corresponding mask images. To the best of our knowledge, this is the first publicly available synthetic polyp dataset and the corresponding generative functions as a PyPI package.

## Materials and methods

In the SinGAN-Seg pipeline, there are as depicted in Fig 1 two main steps: (1) training novel SinGAN-Seg generative models per image and (2) style transfer per image. The first step generates synthetic polyp images and the corresponding binary segmentation masks representing the polyp area. Our method, which is based on the vanilla SinGAN architecture [56], can generate multiple synthetic images and masks from a single real image and the corresponding mask. Therefore, this generation process can be identified as an 1:$N$ generation process. Fig 1 represents this 1:$N$ generation using $[img]_N$, where $N$ represents the number of samples generated using our model and from a real image $[img]$. Then, we apply this step for every image in a target dataset, for which we want to generate synthetic data. The second step focuses on transferring styles such as features of polyps' texture from real images into the corresponding generated synthetic images. This second step is depicted in the Step 2 in Fig 1. This second step is also applied per image.

SinGAN-Seg is a modified version of SinGAN [56], which was designed to generate synthetic data from a GAN trained using only a single image. The original SinGAN is trained using different scales (from 0 to 9) of the same input image, the so-called image pyramid. This image pyramid is a set of images of different resolutions of a single image from low resolution to high resolution. SinGAN consists of a GAN pyramid to train using the corresponding image pyramid. In our study, we build on the implementation and the training process used in SinGAN, except for the number of input and output channels. The original SinGAN implementation [56] uses a three-channel RGB image as input and produces a three-channel RGB image as output. However, our SinGAN-Seg uses four-channel images as the input and the output. The four-channel image consists of the three-channel RGB image and the single-channel ground truth segmentation mask by stacking them together as depicted in the SinGAN-Seg model in Fig 1. The main purpose of this modification is to generate four-channel synthetic output, which consists of a synthetic image and the corresponding segmentation mask. We have done internal modifications to the vanilla SinGAN to handle four-channel input and output.

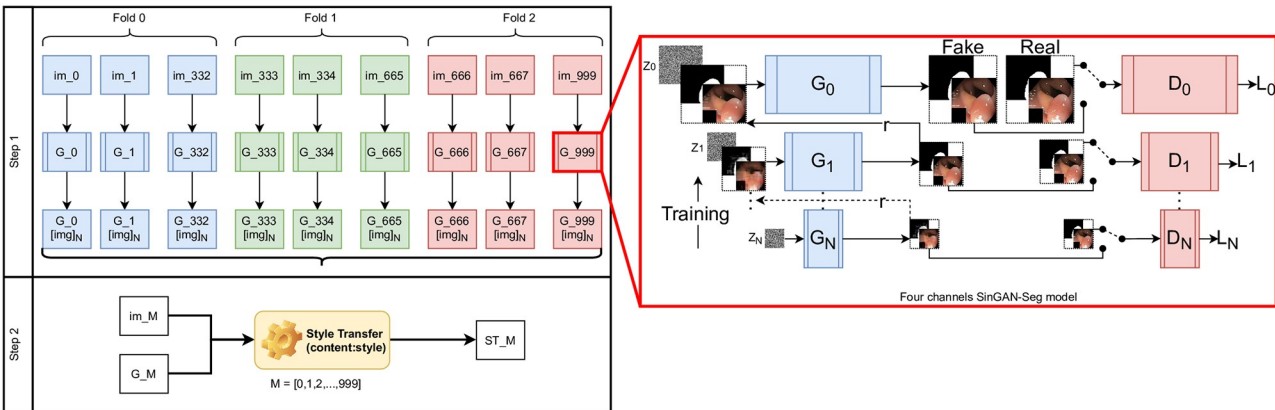

**Fig 1. The complete pipeline of SinGAN-Seg to generate synthetic segmentation datasets.** *Step 1* represents the training of four channels SinGAN-Seg models. *Step 2* represents a fine-tuning step using the neural style transfer [57]. The *four channels SinGAN* is a single training step of our model. Note the stacked input and output compared to the original SinGAN implementation [56] which input only a single image with a noise vector and output only an image. In our SinGAN implementation, all the generators (from $G_0$ to $G_{N-1}$), except $G_N$, get four channels image (a polyp image and a ground truth) as the input in addition to the input noise vector. The first generator, $G_N$ get only the noise vector as the input. The discriminators also get four channels images which consist of an RGB polyp image and a binary mask as input. The inputs to the discriminators can be either real or fake.

In the second step of the SinGAN-Seg pipeline, we fine-tune the output of the four-channel SinGAN-Seg model using the style-transfer method introduced by [57]. This method is also called Neural-style or Neural-transfer, which can take an image and reproduce a new image with a new artistic style. This algorithm calculates two distances, the content distance ($D_C$) and the style distance ($D_S$) to the third image. In the training process of this algorithm, contents and styles are transferred to the third image using the *content*: *style* ratio. More information about this algorithm can be found in the original paper [57]. Using this style transfer algorithm, we aim to improve the quality of the generated synthetic data by transferring realistic styles from real images to synthetic images. As depicted in Step 2 in Fig 1, every generated image $G_M$ is enhanced by transferring style form the corresponding real image $im_M$. Then, the style transferred output image is presented using $ST_M$ where $M = [0, 1, 2\ldots999]$ in this study, representing the 1, 000 images in the training dataset. In this process, a suitable *content*: *style* ratio should be found, and it is a hyper-parameter in this second stage. However, this step is a separate training step from the training step of the SinGAN-Seg generative models. Therefore, this step is optional to follow, but we strongly recommend this style-transferring step to enhance the quality of the output data from the first step.

## Experiments and results

This section presents the experiments and results from our experiments using a polyp dataset [51] as a case study. For all the experiments discussed in the following sections, we have used the Pytorch deep learning framework [58].

### Data

We have used a polyp dataset published with HyperKvasir dataset [51], which consists of polyp findings extracted from endoscopy examinations. HyperKvasir contains 1, 000 polyp images with corresponding segmentation masks annotated by medical experts. We use only this polyp dataset as a case study because of the time and resource-consuming training process

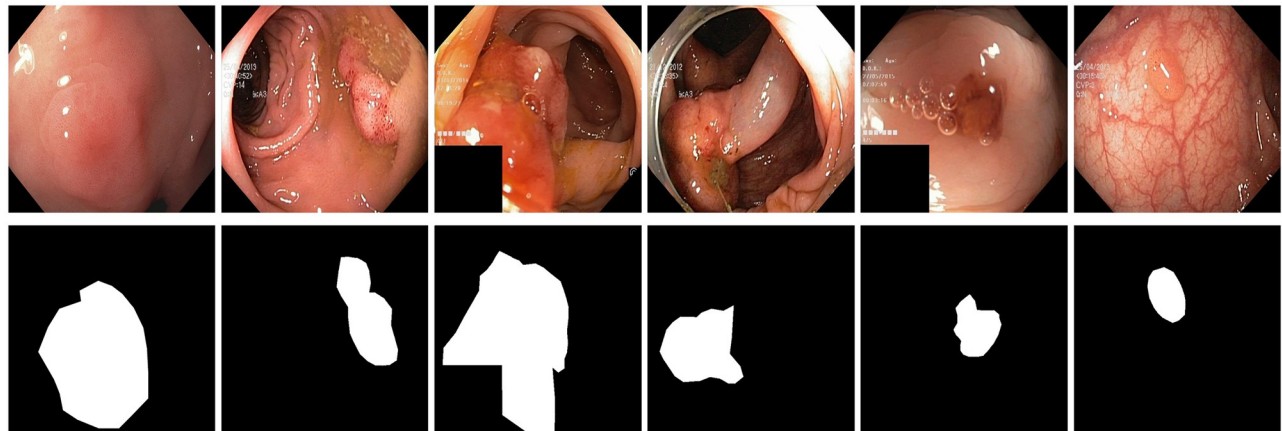

**Fig 2. Sample images and corresponding segmentation masks from HyperKvasir [51].**

of the SinGAN-Seg pipeline. However, the SinGAN-Seg model and pipeline can be used for any segmentation dataset.

A few sample images and the corresponding masks of the polyp dataset in HyperKvasir are shown in Fig 2. The polyp images are RGB images. The masks of the polyp images are single-channel images with white (255) for true pixels, which represent polyp regions, and black (0) for false pixels, which represent clean colon or background regions. In this dataset, there are different sizes of polyps. The distribution of polyp sizes as a percentage of the full image size is presented in the histogram plot in Fig 3, and we can observe that there are more relatively small polyps compared to larger polyps. Additionally, a subset of this dataset was used to prove that the performance of segmentation models trained with small datasets can be improved using our SinGAN-Seg pipeline, and the whole dataset was used to show the effect of using SinGAN-Seg generated synthetic images instead of a large dataset which has enough data to train segmentation models. In this regard, this dataset was used for two purposes:

1. To train SinGAN-Seg models to generate synthetic data.

2. To compare the performance of segmentation ML models trained using both real and synthetic data.

### Training generators

To use SinGAN-Seg to generate synthetic segmentation datasets to represent real segmentation datasets, we first trained SinGAN-Seg models one by one for each image in the training dataset. In our case study, there were 1, 000 polyp images and corresponding ground truth masks. Therefore, 1, 000 SinGAN-Seg models were trained for each image because our models are trained using a single image. This is a time-consuming process, but we can use these pre-trained models repeatedly to generate unlimited number of synthetic data samples. To train these SinGAN-Seg models, we have followed the same SinGAN settings used in the initial paper [56]. Despite using the original training process, the input and output of SinGAN-Seg are four channels. After training each SinGAN-Seg by iterating 2, 000 epochs per scale of pyramidal GAN structure (see the four channels SinGAN-Seg architecture in Fig 1 to understand this pyramidal GAN structure), we stored final checkpoints to generate synthetic data in the later stages from each scale. The resolution of the training image of the SinGAN-Seg model is

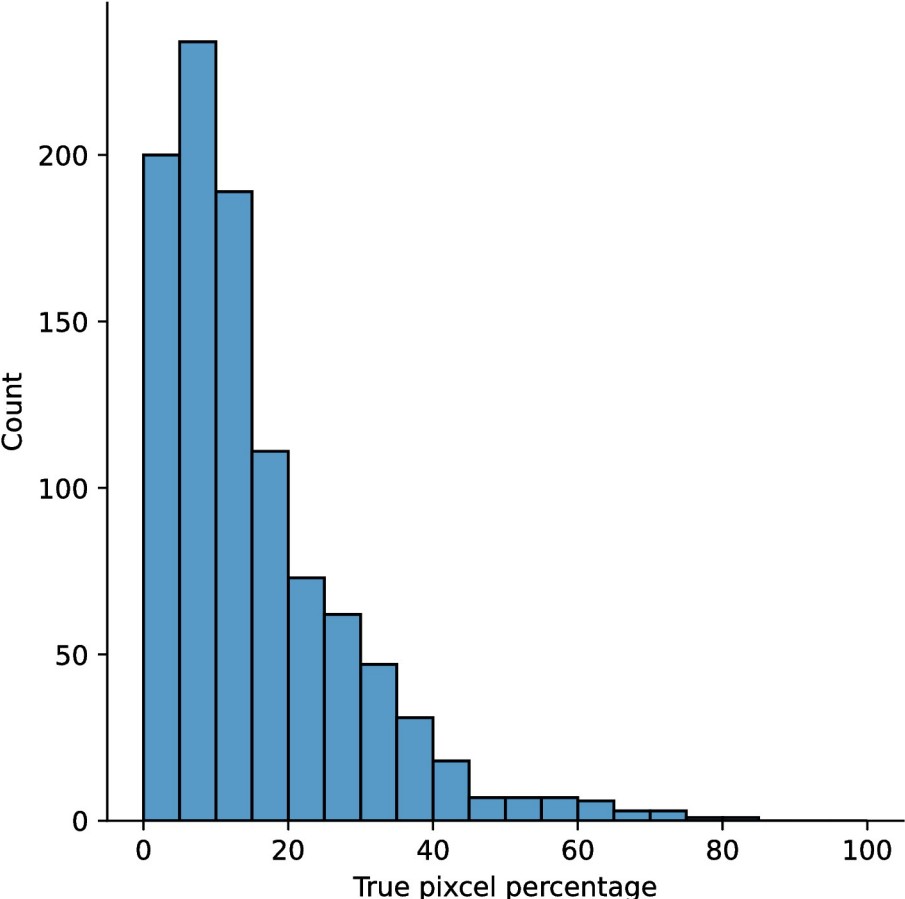

**Fig 3. Distribution of true pixel percentages of the full image size of polyp masks in HyperKvasir [51].**

arbitrary because it depends on the size of the real polyp image. This input image is resized according to the pyramidal rescaling structure introduced in the original implementation of SinGAN [56]. The rescaling pattern is depicted in the four channels SinGAN architecture in Fig 1. The rescaling pattern used to train SinGAN-Seg models is used to change the randomness of synthetic data when pre-trained models are used to generate synthetic data. The models were trained on multiple computing nodes such as Google Colab with Tesla P100 16GB GPUs and a DGX-2 GPU server with 16 V100 GPUs because training 1, 000 GAN architectures one by one is a tremendous task. The average training time per SinGAN-Seg model was around 65 minutes.

After training SinGAN-Seg models, we generated 10 random samples per real image using the input scale 0, which is the lowest scale that uses a random noise input instead of a re-scaled input image. For more details about these scaling numbers and corresponding output behaviors, please refer to the vanilla SinGAN paper [56]. Three randomly selected training images and the corresponding first 5 synthetic images generated using scale 0 are depicted in Fig 4. The first column of the figure represents the real images and the ground truth mask annotated from experts. The rest of the columns represent randomly generated synthetic images and the corresponding generated mask.

In total, we have generated 10, 000 synthetic polyp images and the corresponding masks. SinGAN-Seg generates random samples with high variations when the input scale is 0. This

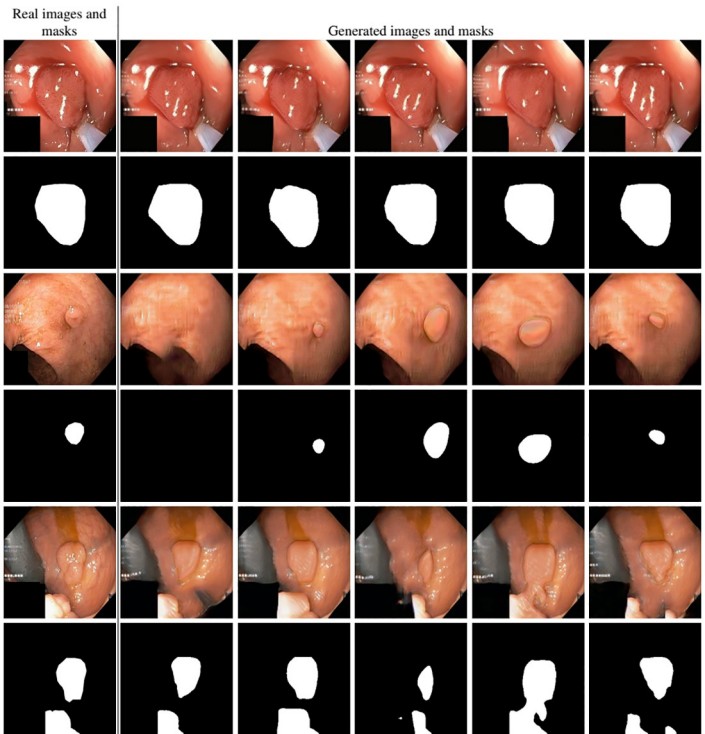

**Fig 4. Real vs. synthetic data.** Samples of real images and corresponding SinGAN-Seg generated synthetic GI-tract images with corresponding masks. The first column contains real images and masks. All other columns represent randomly generated synthetic data from our SinGAN-Seg models, which were trained using the image in the first column.

variation can be easily recognized using the standard deviation (SD) and the mean mask images presented in Fig 5. The mean and SD images were calculated by stacking the 10 generated mask images corresponding to the 10 synthetic images related to a real image and calculating pixel-wise std and mean. Bright color in std images and dark color in mean images mean low variance of pixels. In contrast, dark color in std and bright color in mean images reflect high variance in pixel values. By investigating Fig 5, we see that small polyp masks have high variance compared to the large polyp mask as presented in the figure.

To understand the difference between the mask distribution of real images and synthetic images, we plotted pixel distribution of masks of synthetic 10, 000 images in Fig 6. This plot is comparable to the pixel distribution presented in Fig 3. The randomness of the generations made differences in the distribution of true pixel percentages compared to the true pixel distribution of real masks of real images. However, the overall shape of synthetic data mask distribution shows a more or less similar distribution pattern to the real true pixel percentage distribution.

## Style transferring

After finishing the training of 1, 000 SinGAN-Seg models, the style transfer algorithm [57] was applied to every synthetic sample generated from SinGAN-Seg. In the style-transferring algorithm, we can change several parameters such as the number of epochs to transfer style from one image to another and the *content*: *style* weight ratio. In this paper, we used 1, 000 epochs

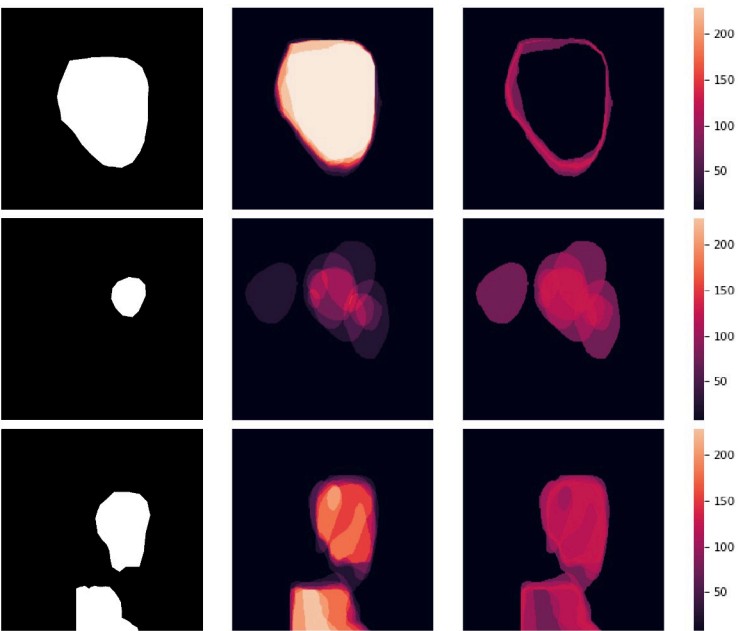

**Fig 5. Analyzing diversity of generated data.** Real masks are presented on left column. Mean(middle column) and standard deviation (right column) calculated from 10 random masks generated from SinGAN-Seg.

to transfer style from a style image (real polyp image) to a content image (generated synthetic polyp). For performance comparisons, two *content*: *style* ratios, i.e., 1: 1 and 1: 1, 000, were used. An NVIDIA GeForce RTX 3080 GPU took around 20 seconds to transfer the style for a single image.

In Fig 7, we provide a visual comparison between pure generated synthetic images and style transferred images (*content*: *style* = 1: 1, 000). Samples with the style transfer ratio 1: 1 are not depicted here because it is difficult to see the differences visually. The first column of Fig 7 shows the real images used as content images to transfer styles. The rest of the images in the first row of each image shows synthetic images generated from SinGAN-Seg before applying the style transferring algorithm. Then, the images in the second row in the figure show the style transferred synthetic images. Differences of the synthetic images before and after applying the style transfer method can be easily recognized from images of the second reference image (using $3^{rd}$ and $4^{th}$ rows in Fig 7).

In addition to the visual comparison, we have calculated single image fréechet inception distance (SIFID), which was introduced in [56], between the real polyp dataset and generated synthetic datasets from our model. These SIFID values are shown in Table 1 with the mean values and SD calculated with five different synthetic datasets from each category, such as without style transferred and with style transferred of 1: 1 and 1: 1, 000 *content*: *style* ratios. The low mean-SIFID value of 0.2216 after applying the style transfer method as the post-processing technique shows the importance of this style-transfer method.

Furthermore, the 1: 1, 000 style transfer ratio shows a slight improvement over the 1: 1 ratio. Therefore, we have selected both ratios to generate synthetic data for the segmentation experiments. Preliminary experiments for other ratios such as 1: 10 and 1: 1, 000, 000 show that we cannot neglect these ratios when applying the style transfer mechanisms while we can see little improvement with high ratios.

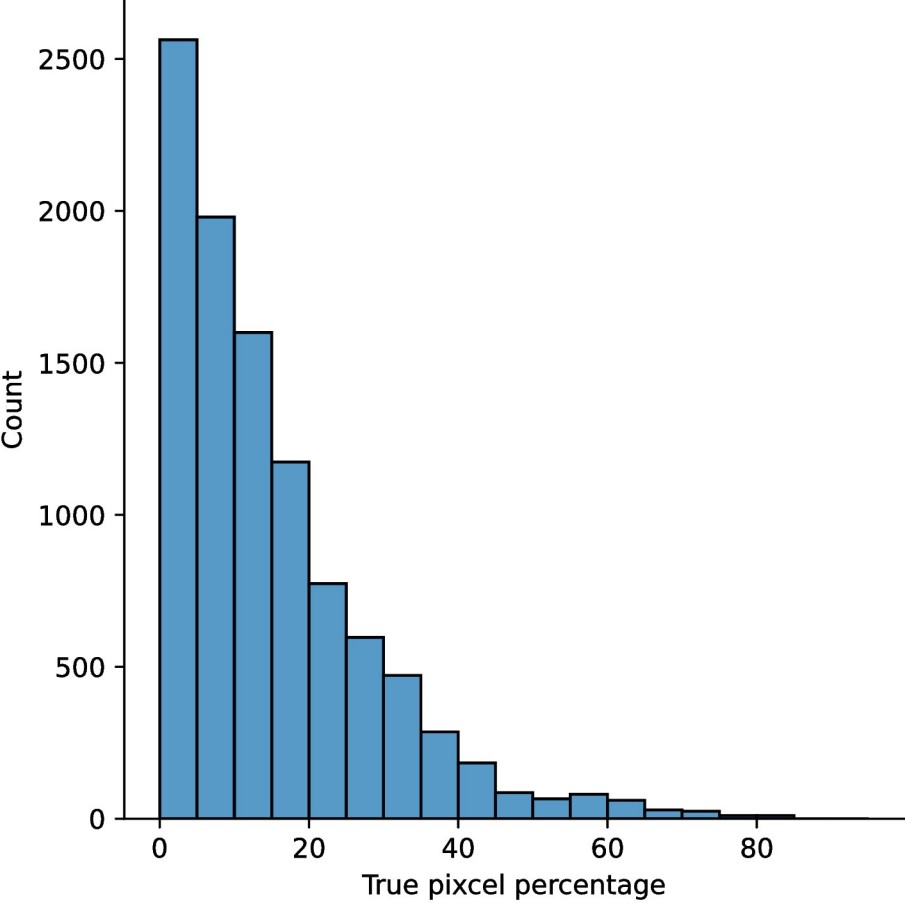

**Fig 6. Distribution of the true pixel percentages calculated from 10, 000 synthetic masks generated with synthetic images using SinGAN-Seg.** The 10, 000 generated images represent the 1, 000 real polyp images. From each real image, 10 synthetic samples were generated. The synthetic 10, 000 dataset can be downloaded from https://osf.io/xrgz8/.

In conclusion, we see a higher qualitative output after applying the style transfer algorithm from real images to generated images. Based on the SIFID values in Table 1, we found that 1: 1, 000 *content*: *style* ratio is better than 1: 1. Well-performing style transfer ratios can be obtained by calculating the SIFID values between the real datasets and the synthetic dataset after applying style transfer.

## Baseline experiments

Two different sets of baseline experiments were performed for two different objectives. The first objective was to compare the quality of generated synthetic data over the real data. Using these baseline experiments, we can identify the capability of sharing SinGAN-Seg synthetic data instead of the real datasets to make is easier to share them between health professional. The second objective was to test how to use SinGAN-Seg pipeline to improve the segmentation performance when the size of the training dataset of real images and masks is small. For all the baseline experiments, we selected UNet++ [59] as the main segmentation model according to the performance comparison done by the winning team at EndoCV 2021 [47]. The single-channel Dice loss function used in the same study was used to train the UNet++ polyp

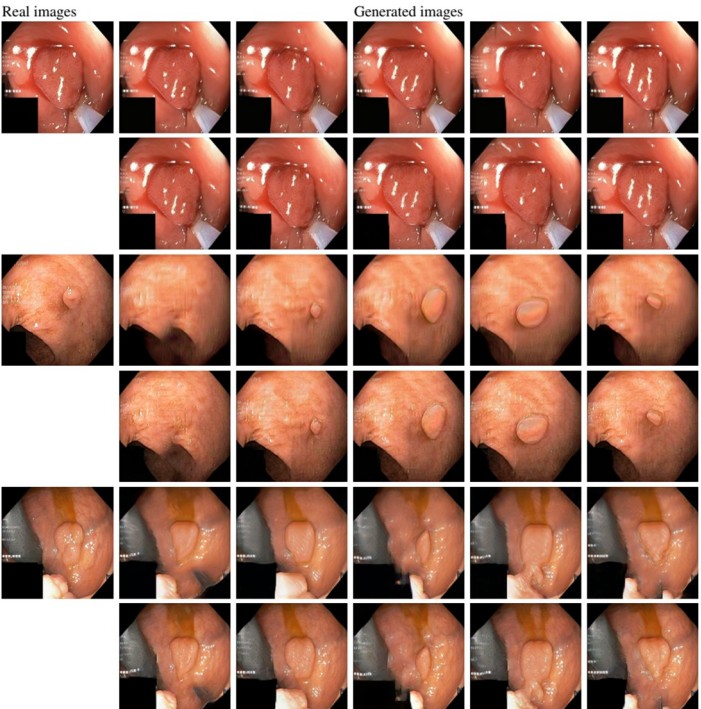

**Fig 7. Direct generations of SinGAN-Seg versus style transferred samples.** The style transferring was performed using 1: 1, 000 content to style ratio. The first row of generated images present quality of images before applying the style transferring and the second row of the same image shows images after applying style transferring. It can be observed that the second row with the style transferring gives better quality.

segmentation models. Moreover, we used the `se_resnext50_32x4d` network as the encoder of the UNet++ model and `softmax2d` as the activation function of the last layer, according to the result of the winning team at EndoCV 2021 [47].

The Pytorch deep learning library was used as the main development framework for the baseline experiments. The training data stream was handled using the PYRA [45] data loader with the Albumentations augmentation library [60]. The real and synthetic images were resized into 128 × 128 using this data handler for all the baseline experiments to save training time. We have used an initial learning rate of 0.0001 for 50 epochs, and then we changed it to 0.00001 for the rest of the training epochs for all the training processes of UNet++. The UNet++ models used to compare real versus synthetic data were trained 300 epochs in total. On the other hand, the UNet++ models used to measure the effect of using SinGAN-Seg synthetic

**Table 1. SIFID value comparison for real versus fake images generated from the SinGAN-Seg models.**

| Target dataset (1k images) | Set 1 | Set 2 | Set 3 | Set 4 | Set 5 | Mean | SD |
|---|---|---|---|---|---|---|---|
| Real | -1.31E-14 | | | | | | - |
| SinGAN-Seg (No ST) | 0.3515 | 0.3499 | 0.3527 | 0.3466 | 0.3480 | 0.3497 | 0.0025 |
| SinGAN-Seg (ST-1:1) | 0.2218 | 0.2214 | 0.2217 | 0.2216 | 0.2215 | 0.2216 | 0.0001 |
| SinGAN-Seg (ST-1:1000) | 0.2206 | 0.2202 | 0.2204 | 0.2204 | 0.2203 | 0.2204 | 0.0001 |

All the SIFID values are calculated w.r.t the real dataset which has 1,000 polyp images. The baseline value calculated between the real dataset and the same real dataset is presented in the first row. The best mean SIFID value is represented using bold text. ST: style transfer, SD: standard deviation.

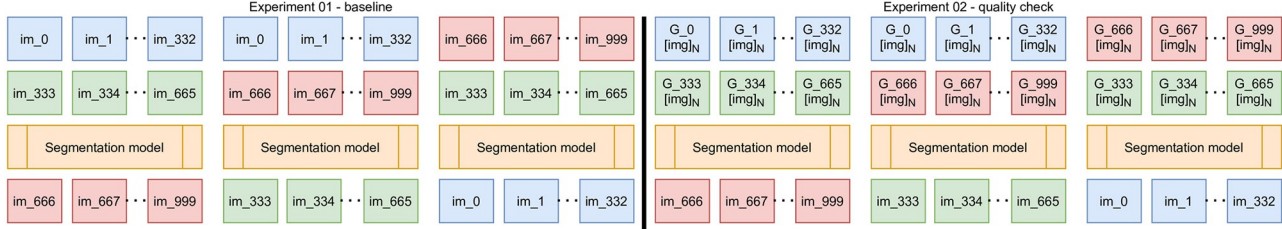

**Fig 8. The experiment setup to analyze the quality of SinGAN output.** Experiment 01—the baseline experiments performed only using the real data. Experiment 02—in this experiment, generated synthetic data is used to train segmentation models, and the real data is used to measure the performance metrics.

data for small segmentation datasets were trained using only 100 epochs because the size of the data splits used to train the models are getting bigger when increasing the training data. In all the experiments, we have selected the best checkpoint using the best validation intersection over union (IOU) score. Finally, Dice loss, IOU score, F-score, accuracy, recall, and precision were calculated for comparisons using validation folds. More details about these evaluation metrics can be found in [61].

**Synthetic data versus real data for segmentation.** We have performed three-fold cross-validation to compare the polyp segmentation performance using UNet++ when using either real or synthetic data for training. First, we divided the real dataset (1, 000 polyp images and the corresponding segmentation masks) into three folds. Then, the trained SinGAN-Seg generative models and the corresponding generated synthetic data were also divided into the same three folds. These three folds are presented using three colors in Step I of Fig 1. In the other experiments, training data and synthetic data folds were not mixed with the validation data folds. If mixed, it leads to a data leakage problem [62].

Then, the baseline performance of the UNet++ model was evaluated using the three folds of the real data. In this experiment, the UNet++ model was trained using two folds and validated using the remaining fold of the real data. In total, three UNet++ models were trained and calculated the average performance using Dice loss, IOU score, F-score, accuracy, recall, and precision only for the polyp class because the most important class of this dataset is the polyp class. This three-fold baseline experiment setup is depicted on the left side of Fig 8.

The usability of synthetic images and corresponding masks generated from SinGAN-Seg was investigated using three-fold experiments as organized in the right side of Fig 8. In this case, UNet++ models were trained only using synthetic data generated from pre-trained generative models and tested using the real data folds, which were not used to train the generative models used to generate the synthetic data. Five different $N(N = [1, 2, 3, 4, 5])$ amount of synthetic data were generated per image to train UNet++ models. This data organization process can be identified easily using the color scheme of the figure. To test the quality of pure generations, we first used the direct output from SinGAN-Seg to train the UNet++ models. Then, the style transfer method was applied with 1: 1 *content*: *style* ratio for all the synthetic data. These style transferred images were used as training data and tested using the real dataset. In addition to the 1: 1 ratio, 1: 1, 000 was tested as a style transfer ratio for the same set of experiments.

Table 2 shows the results collected from the UNet++ segmentation experiments for the baseline experiment and the experiments conducted with synthetic data, which contains pure generated synthetic data and style transferred data using 1: 1 and 1: 1, 000. Differences in IOU scores of these three experiments are plotted in Fig 9 for easy comparison.

**Table 2. Three-fold average of basic metrics to compare real versus synthetic performance with UNet++ and the effect of style-transfers performance.**

| Train data | ST (cw:sw) | Dice loss | IoU | f-score | Accuracy | Recall | Precision |
|---|---|---|---|---|---|---|---|
| **REAL** | NA | **0.1123** | **0.8266** | **0.8882** | **0.9671** | **0.8982** | **0.9161** |
| FAKE-1 | No ST | 0.1645 | 0.7617 | 0.8357 | 0.9531 | 0.8630 | 0.8793 |
| | 1:1 | 0.1504 | 0.7782 | 0.8500 | 0.9572 | 0.8672 | 0.8917 |
| | 1:1000 | 0.1473 | 0.7820 | 0.8530 | 0.9591 | 0.8624 | 0.9005 |
| FAKE-2 | No ST | 0.1549 | 0.7729 | 0.8453 | 0.9561 | 0.8692 | 0.8895 |
| | 1:1 | 0.1550 | 0.7765 | 0.8453 | 0.9575 | **0.8729** | 0.8852 |
| | 1:1000 | 0.1477 | 0.7854 | 0.8525 | 0.9609 | 0.8647 | 0.9038 |
| FAKE-3 | No ST | 0.1610 | 0.7683 | 0.8391 | 0.9556 | 0.8568 | 0.8945 |
| | 1:1 | 0.1475 | 0.7845 | 0.8525 | 0.9585 | 0.8723 | 0.8936 |
| | 1:1000 | 0.1408 | 0.7923 | 0.8593 | 0.9629 | 0.8693 | 0.9078 |
| FAKE-4 | No ST | 0.1649 | 0.7638 | 0.8352 | 0.9525 | 0.8669 | 0.8780 |
| | 1:1 | 0.1464 | 0.7848 | 0.8537 | 0.9594 | 0.8713 | 0.8921 |
| | 1:1000 | **0.1370** | **0.7983** | **0.8630** | **0.9636** | 0.8653 | 0.9185 |
| FAKE-5 | No ST | 0.1654 | 0.7668 | 0.8345 | 0.9563 | 0.8565 | 0.8919 |
| | 1:1 | 0.1453 | 0.7887 | 0.8547 | 0.9610 | 0.8703 | 0.9000 |
| | 1:1000 | 0.1458 | 0.7889 | 0.8543 | 0.9620 | 0.8527 | **0.9211** |

*cw*: *sw* is the short form of *content*: *style* weights ratio. Baseline performance values of using real data are represented using bold text. Moreover, the best performance values of using fake data are marked using bold text. FAKE-N represents a number of images generated from a single image using our model, i.e., FAKE-1 to represent one fake image, FAKE-2 to represent two fake images per real image, etc.

**Synthetic segmentation data generation using few real data samples.** The main purpose of these experiments is to find the effect of using synthetic data generated from the SinGAN-Seg pipeline instead of small real datasets because the SinGAN-Seg pipeline can generate an unlimited number of synthetic samples per real image. A synthetic sample consists of a synthetic image and the corresponding ground truth mask. Therefore, experts' knowledge is not required to annotate the ground truth mask. For these experiments, we have selected the best parameters of the SinGAN-Seg pipeline from the experiments performed under Section. First, we created 10 small sub-datasets from the real polyp images from fold one such that each dataset contains $R$ number of images, where $R$ can be one of the values of [5, 10, 15, ..., 50]. The corresponding synthetic dataset was created by generating 10 synthetic images and corresponding masks per real image. Then, our synthetic datasets consist of $S$ number of images such that $S = $ [50, 100, 150, ..., 500]. Then, we have compared true pixel percentages of real masks and synthetic masks generated from the SinGAN-Seg pipeline using histograms of bin size of 5. The histograms are depicted in Fig 10. The first row represents the histograms of real small detests, and the second row represents the histograms of corresponding synthetic datasets. Compare pairs (one from the top row and the corresponding one from the bottom) to get a clear idea of how the generated synthetic data improved the distribution of masks.

The UNet++ segmentation models were trained using these real and synthetic datasets separately. The synthetic dataset is generated using style transfer ratio 1: 1, 000 because it shows the best performance in the experiment, which uses only fake data to train segmentation models as presented in Table 2 in addition to the best SIFID values presented in Table 1. Then, we have compared the performance differences using validation folds. In these experiments, the training datasets were prepared using fold one. The remaining two folds were used as validation datasets. The collected results from the UNet++ models trained with the real datasets and

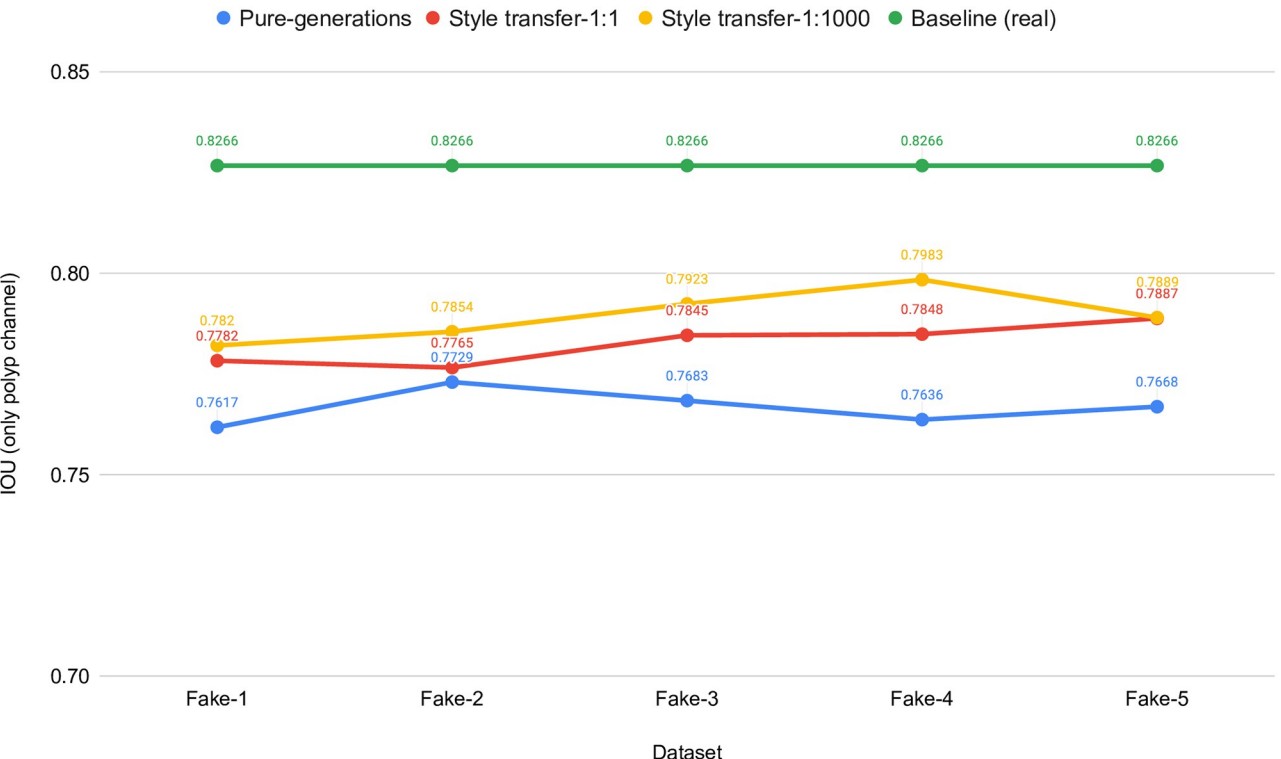

**Fig 9. Real versus synthetic data performance comparison with UNet++ and the effect of applying the style-transferring post processing.** Note that Y-axis starts from 0.70 for better visualization of differences.

the synthetic datasets are tabulated in Table 3. A comparison of the corresponding IOU score scores are plotted in Fig 11.

In this experiment, we have evaluated how synthetic data can be used instead of small real datasets, such as 5 − 50 images per dataset. Results collected from these experiments show that synthetic segmentation datasets can produce better segmentation performance when the corresponding real datasets are small. For the smallest dataset, the performance gain is around 30% in term of IOU score.

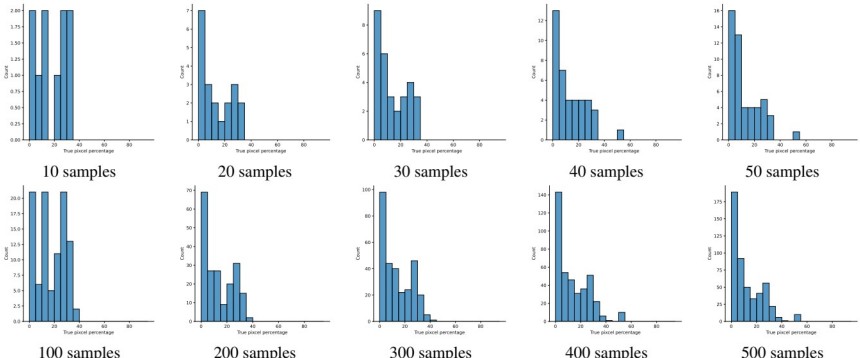

**Fig 10. Distribution comparison between real (top row) and synthetic (bottom row) masks.** Synthetic masks were generated using the SinGAN-Seg.

**Table 3. Real versus fake comparisons for small datasets after applying the style transfer method with a 1: 1000 ratio for fake data.**

| | # | Dice loss | IoU | f-score | Accuracy | Recall | Precision |
|---|---|---|---|---|---|---|---|
| Real | 5 | 0.4662 | 0.4618 | 0.5944 | 0.8751 | 0.7239 | 0.6305 |
| Fake | 50 | **0.3063** | **0.5993** | **0.7048** | **0.9211** | **0.7090** | **0.8133** |
| Real | 10 | 0.3932 | 0.5969 | 0.7079 | 0.9164 | 0.7785 | 0.7516 |
| Fake | 100 | **0.2565** | **0.6478** | **0.7457** | **0.9259** | **0.7911** | **0.7970** |
| Real | 15 | 0.2992 | 0.6431 | 0.7402 | 0.9322 | 0.7388 | **0.8602** |
| Fake | 150 | **0.2852** | **0.6559** | **0.7624** | **0.9329** | **0.8172** | 0.7833 |
| Real | 20 | 0.3070 | **0.6680** | **0.7668** | 0.9328 | **0.7771** | 0.8566 |
| Fake | 200 | **0.2532** | 0.6569 | 0.7544 | **0.9342** | 0.7317 | **0.8827** |
| Real | 25 | **0.2166** | **0.6995** | **0.7929** | 0.9405 | **0.7955** | 0.8804 |
| Fake | 250 | 0.2182 | 0.6961 | 0.7860 | **0.9418** | 0.7690 | **0.8957** |
| Real | 30 | **0.2100** | **0.7037** | **0.7971** | **0.9417** | **0.8005** | 0.8758 |
| Fake | 300 | 0.2228 | 0.6843 | 0.7797 | 0.9388 | 0.7683 | **0.8810** |
| Real | 35 | **0.2164** | **0.6955** | **0.7889** | **0.9398** | **0.8157** | 0.8456 |
| Fake | 350 | 0.2465 | 0.6677 | 0.7543 | 0.9346 | 0.7385 | **0.8933** |
| Real | 40 | **0.2065** | **0.7085** | **0.7974** | **0.9417** | 0.7881 | **0.8947** |
| Fake | 400 | 0.2194 | 0.6894 | 0.7816 | 0.9305 | **0.8276** | 0.8219 |
| Real | 45 | **0.1982** | **0.7188** | **0.8062** | **0.9441** | **0.8120** | **0.8839** |
| Fake | 450 | 0.2319 | 0.6794 | 0.7697 | 0.9341 | 0.7859 | 0.8633 |
| Real | 50 | **0.2091** | **0.7115** | **0.7948** | **0.9418** | 0.7898 | **0.8932** |
| Fake | 500 | 0.2255 | 0.6896 | 0.7756 | 0.9380 | **0.7961** | 0.8644 |

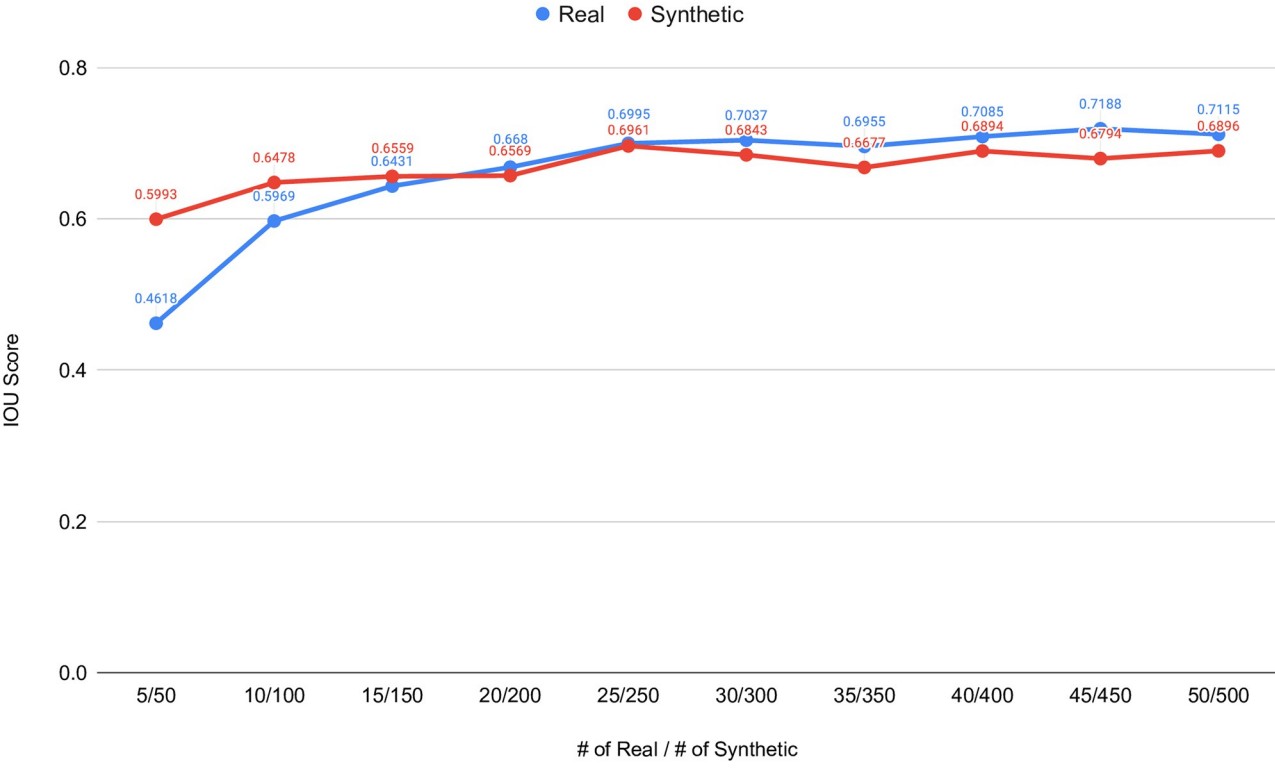

**Fig 11. Real versus fake performance comparison with small training datasets.** Fake datasets are generated with the style transfer method using *content*: *style* ratio of 1: 1, 000.

**Table 4. FID value comparison between the real dataset of 1000 real images and the synthetic datasets of 1000 synthetic images generated from different GAN architectures which are modified to generate four channels outputs.**

| GAN architecture | Set 1 | Set 2 | Set 3 | Set 4 | Set 5 | Mean | SD |
|---|---|---|---|---|---|---|---|
| DCGAN [63] | 270.82 | 269.79 | 268.38 | 268.32 | 269.13 | 269.29 | 1.05 |
| Progressive GAN [64] | 285.81 | 284.30 | 282.81 | 283.54 | 285.00 | 284.29 | 1.18 |
| FastGAN [65] | 74.60 | 74.43 | 75.53 | 75.08 | 76.20 | 75.17 | 0.72 |
| SinGAN-Seg (ours) | 99.61 | 98.12 | 98.27 | 97.59 | 97.86 | 98.29 | 0.78 |
| SinGAN-Seg with Style transfer (ours) | **43.74** | **43.35** | **43.71** | **43.41** | **43.11** | **43.46** | **0.26** |

The all GAN architectures were trained with the whole polyp dataset which has 1000 polyp images and the corresponding ground-truth masks. Then, 1000 synthetic images were generated using the best checkpoints of each GAN models. Style transfer ratio used in SinGAN-Seg is 1: 1000. The best values are presented using **bold** text.

**SinGAN-Seg versus state-of-the-art deep generative models.** In this section, we analyze the effect of using other state-of-the-art GANs to generate synthetic data and corresponding masks. To perform this analysis, we selected three GAN architectures, namely DCGAN [63], Progressive GAN [64] and FastGAN [65], which represent simple to advanced deep generative models and can be trained with small datasets in contrast to other state-of-the-art GAN architectures. All the GAN models were modified to generate four-channel output to get the RGB synthetic image and corresponding mask on the fourth channel. All the GAN models were trained using the recommended hyper-parameters and the number of epochs discussed in the original papers.

We used Fréchet inception distance (FID) [66] to compare the output of the selected GAN architectures to our SinGAN-Seg model because SIFID is not applicable for comparing different GANs that generate random images compared to one-to-one generation process of SinGAN-Seg. However, a recent study [67] shows that FID depends on the images compression type, image size, and many other factors used to calculate FID values. Therefore, we have used the standard FID calculation method introduced in the recent study [67].

We generated five sample datasets from each GAN architecture to have 1, 000 images per dataset. Then, FID values were calculated between the synthetic datasets and the real datasets. Mean and SD values were calculated using the FID values each sample datasets, where a comparison is shown in Table 4. Furthermore, sample images from this each GAN architecture are presented in Fig 12. Only SinGAN-Seg has two different output. One is before applying style transfer and one is after applying style transfer. Our style transfer mechanism works when we can find one-to-one mapping from real to synthetic. Therefore, this post-processing technique is applicable only for SinGAN-Seg.

According to the FID values and images in Fig 12, it is clear that FastGAN and SinGAN-Seg versions generate better quality synthetic images compared to DCGAN and Progressive GAN. FastGAN shows better FID scores than SinGAN-Seg without style transfer. However, SinGAN-Seg-ST, which is SinGAN-Seg with style transfer shows the best result among all the GANs is when 1, 000 images were available for training.

As FastGAN showed the best performance among all the other GANs is, we decided to compare it with SinGAN-Seg and SinGAN-Seg-ST to evaluate generating synthetic data when training on datasets that contain few samples. We trained FastGAN architecture with a small number of images, such as 5, 10, 15, . . ., 50, and generated the corresponding synthetic datasets, which have synthetic images from 5 to 50. Then, we used the synthetic datasets to compare the performance with FastGAN trained on small datasets. FID values were calculated between the synthetic images and corresponding real images used to train the models. Mean

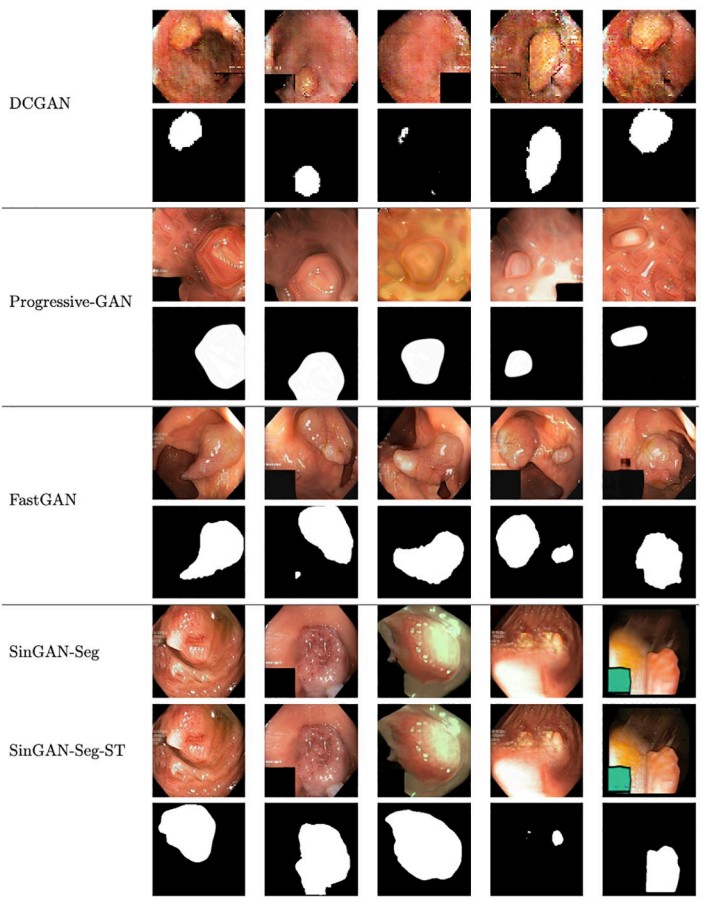

**Fig 12. Sample images generated from different GAN architectures.** SinGAN-Seg has two versions: one is without style transfer (SinGAN-Seg) and one is with style transfer (SinGAN-Seg-ST). The best ratio of *content*: *style* = 1: 1, 000 was used for transferring style.

and standard deviation (SD) of FID values calculated from five synthetic datasets (Set 1, Set 2, Set 3, Set 4 and Set 5) generated from each GAN model are tabulated in Table 5. Moreover, mean FID values of FastGAN, SinGAN-Seg and SinGAN-Seg-ST are compared in Fig 13.

According the the table and the figure, it is clear that FastGAN is unstable with small training datasets while SinGAN-Seg and SinGAN-Seg-ST show good progress toward the number of images used to calculate FID values. SinGAN-Seg-ST is better than SinGAN-Seg as we experienced with previous experiments discussed in this study. It it worth to remind that SinGAN-Seg and SinGAN-Seg-ST does not depend on number of training images because it needs only a single image to train. Therefore, SinGAN-Seg model and it´s training pipeline used in this study prove the importance using it to generate synthetic datasets when GAN models do not have access to large training datasets.

## Discussion

The SinGAN-Seg pipeline has two steps. The first one is generating synthetic polyp images and the corresponding ground truth masks. The second is transferring style from real polyp images to synthetic polyp images to make them more realistic compared to the pure generation of images using only SinGAN-Seg from the first step. We have developed this pipeline to

**Table 5. FID value calculations between real and synthetic datasets generated from FastGAN, SinGAN-Seg and SinGAN-Seg-ST trained with small datasets.**

| GAN type | # of synthetics | Set 1 | Set 2 | Set 3 | Set 4 | Set 5 | Mean | SD |
|---|---|---|---|---|---|---|---|---|
| FastGAN | 5 | 290.92 | 224.22 | 224.10 | 240.78 | 252.58 | 246.52 | 27.57 |
| | 10 | 264.98 | 252.69 | 243.38 | 230.54 | 223.88 | 243.09 | 16.57 |
| | 15 | 235.64 | 231.45 | 213.92 | 220.76 | 244.86 | 229.33 | 12.21 |
| | 20 | 249.57 | 268.03 | 259.91 | 261.15 | 264.05 | 260.54 | 6.88 |
| | 25 | 356.25 | 354.63 | 354.06 | 355.91 | 354.32 | 355.04 | **0.98** |
| | 30 | 312.53 | 299.09 | 298.20 | 302.74 | 298.36 | 302.18 | 6.07 |
| | 35 | 266.21 | 273.29 | 265.75 | 266.97 | 262.26 | 266.90 | 4.01 |
| | 40 | 257.10 | 256.23 | 256.23 | 256.26 | 260.88 | 257.34 | 2.01 |
| | 45 | 228.93 | 226.10 | 226.42 | 222.41 | 219.24 | **224.62** | 3.80 |
| | 50 | 236.34 | 224.25 | 235.53 | 224.69 | 225.59 | 229.28 | 6.10 |
| SinGAN-Seg (ours) | 5 | 245.80 | 264.39 | 271.63 | 280.08 | 288.36 | 270.05 | 16.27 |
| | 10 | 224.76 | 223.50 | 234.41 | 240.93 | 230.56 | 230.83 | 7.17 |
| | 15 | 208.48 | 208.25 | 217.65 | 222.08 | 212.05 | 213.70 | 6.03 |
| | 20 | 210.21 | 220.01 | 224.06 | 222.79 | 217.42 | 218.90 | 5.49 |
| | 25 | 207.81 | 216.58 | 221.12 | 217.13 | 216.05 | 215.74 | 4.86 |
| | 30 | 200.99 | 208.70 | 211.04 | 210.92 | 206.85 | 207.70 | 4.13 |
| | 35 | 201.40 | 206.30 | 209.27 | 206.52 | 207.09 | 206.12 | 2.89 |
| | 40 | 198.33 | 202.14 | 204.32 | 202.65 | 202.51 | 201.99 | **2.21** |
| | 45 | 188.88 | 190.03 | 194.49 | 192.98 | 195.08 | 192.29 | 2.73 |
| | 50 | 186.83 | 187.84 | 192.75 | 191.32 | 192.60 | **190.27** | 2.76 |
| SinGAN-Seg-ST (ours) | 5 | 139.66 | 152.66 | 153.20 | 170.99 | 168.65 | 157.03 | 12.90 |
| | 10 | 131.15 | 131.93 | 127.93 | 140.65 | 140.34 | 134.40 | 5.76 |
| | 15 | 121.27 | 124.40 | 120.90 | 129.80 | 129.41 | 125.16 | 4.29 |
| | 20 | 119.75 | 133.64 | 123.64 | 129.75 | 126.55 | 126.67 | 5.37 |
| | 25 | 118.98 | 134.12 | 123.60 | 125.04 | 126.60 | 125.67 | 5.51 |
| | 30 | 118.95 | 125.96 | 117.96 | 120.81 | 121.73 | 121.08 | 3.11 |
| | 35 | 121.06 | 124.96 | 121.85 | 120.90 | 121.00 | 121.95 | 1.72 |
| | 40 | 117.79 | 119.27 | 119.24 | 115.24 | 118.06 | 117.92 | 1.64 |
| | 45 | 114.19 | 112.88 | 113.11 | 111.95 | 114.76 | 113.38 | 1.11 |
| | 50 | 111.16 | 110.66 | 111.48 | 112.39 | 111.97 | **111.53** | **0.68** |

achieve two goals. The first one is to make it easier to share medical data and generate more annotated data. The second one is to improve the polyp segmentation performance when the size of the training dataset is small by augmenting the training dataset with synthetic images.

## SinGAN-Seg as data sharing technique

The SinGAN-Seg can generate unlimited synthetic data with the corresponding ground truth mask, representing real datasets. This SinGAN-Seg pipeline is applicable for any dataset with segmentation masks, particularly when the dataset is not allowed to share. However, in this study, we applied this pipeline to a public polyp dataset with segmentation masks as a case study. Assuming that the polyp dataset is private, we used this polyp dataset as a proof of concept medical dataset. In this case, we published a PyPI package, `singan-seg-polyp` which can generate an unlimited number of polyp images and corresponding ground truth masks. If the real polyp dataset is restricted to share, then this type of pre-trained models can be used to generate an alternative dataset to represent the real dataset and share. Alternatively, we can

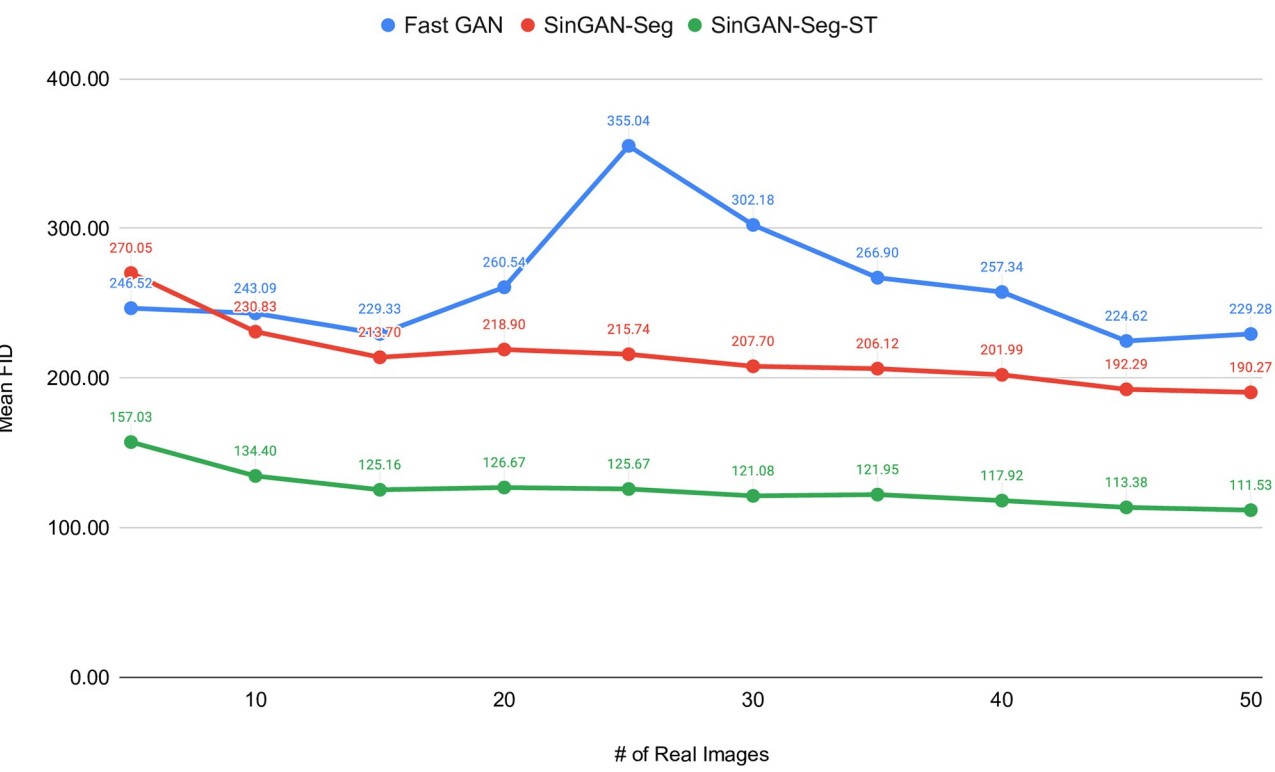

**Fig 13. FastGAN versus SinGAN-Seg and SinGAN-Seg-ST.** SinGAN-Seg-ST represents SinGAN-Seg with style transfer of 1: 1000.

publish a pre-generated synthetic dataset using the SinGAN-Seg pipeline, such as the synthetic polyp dataset published as a case study at https://osf.io/xrgz8.

According to the results presented in Table 2, the UNet++ segmentation network performs slightly better when the real data is used as training data compared to using synthetic data as training data. However, the small performance gap between real and synthetic data as training data implies that the synthetic data generated from the SinGAN-Seg can use as an alternative to sharing segmentation data instead of real datasets, which are restricted to share. The style-transferring step of the SinGAN-Seg pipeline could reduce the performance gap between real and synthetic data as training data for UNet++. The performance gap between real data and the synthetic data as training data for segmentation models is negotiable because the primary purpose of producing the synthetic data is not to improve the performance of segmentation models, but to introduce an alternative data sharing which are practically applicable when datasets should be shared between different health institutions and research labs.

## SinGAN-Seg with small datasets

In addition to using the SinGAN-Seg pipeline as a data-sharing technique when the real data-sets are restricted to publish, the pipeline can improve the performance of segmentation tasks when a dataset is small. In this case, the SinGAN-Seg pipeline can generate synthetic data to overcome the problem associated with the small dataset. In other words, the SinGAN-Seg pipeline acts as a stochastic data augmentation technique due to the randomness of the synthetic data generated from this model. For example, consider a manual segmentation process such as cell segmentation in any medical laboratory experiment. This type of tasks is really

hard to perform for experts as well. As a result, the amount of data collected with manually annotated segmentation masks is limited, and applying only traditional transformations such horizontal flip, shift scale rotation, resizing, motion blur and other augmentation techniques introduced for segmentation task (see Albumentations [60] library for more details) are not enough for achieving good performance. To show that, we have combined SinGAN-Seg with Albumentation data augmentation techniques. Then, SinGAN-Seg shows a performance improvement over the performance achieved only using the traditional augmentation techniques when the initial real dataset size is small (such as 5 to 20).

Our SinGAN-Seg pipeline can increase the size of small datasets, and thus the quality, by generating an unlimited number of random samples from a single manually annotated image. Furthermore, in contrast to our method, conventional GAN models cannot be used to generate high-quality synthetic data when the training datasets have a limited number of images. This is one of the largest advantages of SinGAN-Seg, specially in the medical domain which usually has a lack of good training data. This is proven by training three state-of-the-art conventional GAN models, namely DCGAN, ProgressiveGAN and FastGAN and comparing FID values with SinGAN-Seg models when training datasets consist of a small number of images.

This study showed that the synthetic data generated from a small real dataset can improve the performance of ML models for image segmentation. For example, when the real polyp dataset size is 5 to train our UNnet++ model, the synthetic dataset with 50 samples showed 30% improvement over the IOU score. Similarly, when the real dataset is 10 and the corresponding generated dataset is 100 (we always take 10 times as the real dataset in our case studies, but there is no any limit.), the synthetic dataset shows an 8.5% improvement over the real dataset. These experiments emphasize that SinGAN-Seg generated synthetic data can be used instead of small real datasets.

## Conclusions and future work

This paper presented our four-channel SinGAN-Seg model and the corresponding SinGAN-Seg pipeline with a style transfer method to generate realistic synthetic polyp images and the corresponding ground truth segmentation masks. Our pipeline can be used as an alternative method to provide and share data when real datasets are restricted. Moreover, this pipeline can be used to improve segmentation performance when we have small segmentation datasets, i.e., as an effective data augmentation technique. The results from the conducted three-fold cross-validation segmentation experiments show that models trained on synthetic data can achieve performance very close to the performance of segmentation models using only real data to train the ML models. On the other hand, we also show that SinGAN-Seg can achieve better segmentation performance when the training datasets are very small because of the advantage of being able to learn from one single image. Furthermore, we performed a qualitative and quantitative comparison with other state-of-the-art GANs and show that SinGAN-Seg with style-transfer technique (SinGAN-Seg-ST) performs better than other GAN architectures.

In future studies, researchers can combine super-resolution GAN models [68] with this pipeline to improve the quality of the output after the style transfer step. When we have high-resolution images, ML algorithms show better performance than algorithms trained using low-resolution images [69].

## Code and dataset availability

Using all the pre-trained SinGAN-Seg checkpoints, we have published a PyPI package and the corresponding GitHub repository to make all the experiments reproducible. Additionally, we

have published the first synthetic polyp dataset to demonstrate how to share synthetic data instead of a real dataset. The synthetic dataset is freely available at https://osf.io/xrgz8/ as an example synthetic dataset generated using the SinGAN-Seg pipeline. Furthermore, this dataset is an example showing how to increase a segmentation dataset size without using the time-consuming and costly medical data annotation process that needs medical experts' knowledge.

We named this PyPI package as `singan-seg-polyp` (`pip install singan-seg-polyp`), and it can be found here: https://pypi.org/project/singan-seg-polyp/. To the best of our knowledge, this is the only PyPI package to generate an unlimited number of synthetic polyps and corresponding masks. The corresponding GitHub repository is available at https://github.com/vlbthambawita/singan-seg-polyp. A set of functionalities were introduced in this package for end-users. Generative functions can generate random synthetic polyp data with their corresponding mask for a given image id from 1 to 1, 000 or for the given checkpoint directory, which is downloaded automatically when the generative functions are called. The package contains the style transfer function that can be used to transfer the style from real polyp images to the corresponding synthetic polyp images. In both functionalities, the relevant hyper-parameters can be changed as needed to end-users of this PyPI package.

## Acknowledgments

For this research the Experimental Infrastructure for Exploration of Exascale Computing (eX3), Research Council of Norway Project 270053 was used.

## Author Contributions

**Conceptualization:** Vajira Thambawita, Pegah Salehi, Sajad Amouei Sheshkal, Michael A. Riegler.

**Investigation:** Vajira Thambawita.

**Methodology:** Vajira Thambawita, Pegah Salehi, Sajad Amouei Sheshkal, Michael A. Riegler.

**Validation:** Vajira Thambawita, Pegah Salehi, Sajad Amouei Sheshkal, Steven A. Hicks, Michael A. Riegler.

**Visualization:** Vajira Thambawita, Pegah Salehi, Sajad Amouei Sheshkal.

**Writing – original draft:** Vajira Thambawita, Pegah Salehi, Sajad Amouei Sheshkal, Steven A. Hicks, Hugo L. Hammer, Sravanthi Parasa, Thomas de Lange, Pål Halvorsen, Michael A. Riegler.

**Writing – review & editing:** Vajira Thambawita, Pegah Salehi, Sajad Amouei Sheshkal, Steven A. Hicks, Hugo L. Hammer, Sravanthi Parasa, Thomas de Lange, Pål Halvorsen, Michael A. Riegler.

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
