## [Decision Letter · Decision Letter 0]

10 Dec 2021

PONE-D-21-31343SinGAN-Seg: Synthetic training data generation for medical image segmentationPLOS ONE

Dear Dr. Thambawita,

Thank you for submitting your manuscript to PLOS ONE. After careful consideration, we feel that it has merit but does not fully meet PLOS ONE’s publication criteria as it currently stands. Therefore, we invite you to submit a revised version of the manuscript that addresses the points raised during the review process.

The paper needs significant improvement. It should position the proposed method with respect to the state of the art in GAN-based synthetic data generation - detail the contribution, show the advantages, put the results in comparison. It should also explain the improvement of results due to style transfer. More metrics should be employed to evaluate the generated images.

We look forward to receiving your revised manuscript.

Kind regards,

Ruxandra Stoean

Academic Editor

PLOS ONE

Journal Requirements:

"The research has benefited from the Experimental Infrastructure for Exploration of

Exascale Computing (eX3), which is financially supported by the Research Council of Norway under contract."

Reviewers' comments:

Reviewer's Responses to Questions

**Comments to the Author**

1. Is the manuscript technically sound, and do the data support the conclusions?

Reviewer #1: Partly

Reviewer #2: Partly

2. Has the statistical analysis been performed appropriately and rigorously? 

Reviewer #1: No

Reviewer #2: No

3. Have the authors made all data underlying the findings in their manuscript fully available?

Reviewer #1: Yes

Reviewer #2: Yes

4. Is the manuscript presented in an intelligible fashion and written in standard English?

Reviewer #1: No

Reviewer #2: Yes

5. Review Comments to the Author

Reviewer #1: 1) First of all, the figures need to be significantly improved. Right now they are of very poor quality and very blurred. It is hard for the reviewers to make sense of the figures as they are. Use previously published PLOS ONE papers as a reference.

2) There are many spelling mistakes. Please run the manuscript through a spell check.

3) The proposed approach uses state-of-the-art generation approach SinGAN to generate synthetic images along with the corresponding ground-truth followed by a style transfer. This is not a significant novelty other than the part of generating the corresponding ground-truth.

4) The last 2 contributions of the paper are not new:

a) "We show that synthetic images and corresponding mask images can improve the segmentation performance when the size of a training dataset is limited."

b) "We show that synthetic data can achieve a very close performance to the real data when the real segmentation datasets are large enough."

Recent works such as [1 - 3] have already shown this with medical images. The references are missing.

[1] Theagarajan, R. and Bhanu, B., 2019. DeephESC 2.0: Deep generative multi adversarial networks for improving the classification of hESC. PloS one, 14(3), p.e0212849.

[2] Witmer, A. and Bhanu, B., 2018, October. HESCNET: A Synthetically Pre-Trained Convolutional Neural Network for Human Embryonic Stem Cell Colony Classification. In 2018 25th IEEE International Conference on Image Processing (ICIP) (pp. 2441-2445). IEEE.

[3] Jonnalagedda, P., Weinberg, B., Allen, J., Min, T.L., Bhanu, S. and Bhanu, B., 2021, January. SAGE: Sequential Attribute Generator for Analyzing Glioblastomas using Limited Dataset. In 2020 25th International Conference on Pattern Recognition (ICPR) (pp. 4941-4948). IEEE.

4) quantitative evaluation of the generated synthetic images are required. The authors need to provide scores such as the FID and SIFID scores as used in the SinGAN paper.

4) The results provided in Tables 1 and 2 are not sufficient. Since training data is very important, how are the qualities of the generated images verified in Tables 1 and 2? One suggestion would be to compute the quality metric (FID, SIFID, etc.) of the generated images and take only the top X% of the synthetic images for training

Reviewer #2: The authors have developed a method for generating synthetic data for medical image segmentation based on sinGAN and style transfer. It is an interesting study to deal with privacy and small dataset of medical images. However, there is no comparison with existing methods, and the superiority of the proposed method is not clear. The reviewer suggests the authors to revise the manuscript according to the following comments.

SinGAN-Seg as data sharing technique

1. Please add a comparison of the proposed method with other data augmentation methods (e.g., conventional GAN). What is the advantage of the proposed method over the conventional GAN based synthetic data generation?

SinGAN-Seg with small datasets

2. As mentioned above, please highlight the superiority of the proposed method compared to other common data augmentation methods for small datasets analysis, too.

3. Did you apply style transfer in the results of Table 2 and Fig.11? Please clarify how style transfer improves the results.

6. PLOS authors have the option to publish the peer review history of their article (what does this mean?). If published, this will include your full peer review and any attached files.

Reviewer #1: No

Reviewer #2: No

---

## [Author Response · Author response to Decision Letter 0]

24 Jan 2022

Ruxandra Stoean

Academic editor

PLOS ONE

Dear editor,

Herewith, we submit our revised manuscript titled “SinGAN-Seg: Synthetic training data generation for medical image segmentation”.

Thank you very much for giving us the opportunity to revise our manuscript and resubmit. We really appreciate that the reviewers and you have commented and given us feedback to improve our manuscript. We have thoroughly gone through the reviewers comments and we addressed all comments point by point. The response letters are attached in the submission system for each reviewer.

We believe the review process has improved the quality of our study significantly. Moreover, we have followed the latex format given by PLOS ONE to complete the manuscript and the final tex file is attached in the submission system. 

Regarding the acknowledgement statement, "The research has benefited from the Experimental Infrastructure for Exploration of Exascale Computing (eX3), which is financially supported by the Research Council of Norway under contract." : This is not a funding source for our project, but for a general project which provides the hardware resources to run experiments in our lab. Therefore, the current funding statement "The author(s) received no specific funding for this work." is still valid. 

However, if you think that we should remove the acknowledgement statement then it would be no problem for us to do so. 

Thank you very much.

Yours sincerely,

Vajira Thambawita

vajira@simula.no

---

## [Decision Letter · Decision Letter 1]

9 Feb 2022

PONE-D-21-31343R1SinGAN-Seg: Synthetic training data generation for medical image segmentationPLOS ONE

Dear Dr. Thambawita,

Thank you for submitting your manuscript to PLOS ONE. After careful consideration, we feel that it has merit but does not fully meet PLOS ONE’s publication criteria as it currently stands. Therefore, we invite you to submit a revised version of the manuscript that addresses the points raised during the review process.

Improvement is still needed regarding experimental comparison and formatting.

We look forward to receiving your revised manuscript.

Kind regards,

Ruxandra Stoean

Academic Editor

PLOS ONE

Reviewers' comments:

Reviewer's Responses to Questions

**Comments to the Author**

1. If the authors have adequately addressed your comments raised in a previous round of review and you feel that this manuscript is now acceptable for publication, you may indicate that here to bypass the “Comments to the Author” section, enter your conflict of interest statement in the “Confidential to Editor” section, and submit your "Accept" recommendation.

Reviewer #1: (No Response)

Reviewer #2: All comments have been addressed

2. Is the manuscript technically sound, and do the data support the conclusions?

Reviewer #1: Partly

Reviewer #2: Yes

3. Has the statistical analysis been performed appropriately and rigorously? 

Reviewer #1: No

Reviewer #2: N/A

4. Have the authors made all data underlying the findings in their manuscript fully available?

Reviewer #1: Yes

Reviewer #2: Yes

5. Is the manuscript presented in an intelligible fashion and written in standard English?

Reviewer #1: Yes

Reviewer #2: Yes

6. Review Comments to the Author

Reviewer #1: 1) The authors mentioned that they have uploaded better quality images, but I still do not see any improvement in the quality of the images.

2) There are still some minor spelling mistakes, please check one more time. Line 93 date -> data.

3) Table 1 in the revised manuscript is very poorly formatted. I cannot see anything, except the first column. Please check the manuscript before uploading for review!

4) The authors mention and discuss the quality of the generated images using SinGan-seg throughout the paper but, there is no comparison with other GAN related works. Hence it is not possible to validate if the proposed approach is significantly better in terms of quality, other than the fact that it requires only 1 training image.

4.1) Although SinGan-seg takes only 1 input to train compared to conventional GANs that require more images, the authors need to generate results using at least 2 - 3 other approaches that use conventional GANs (even if it requires more training images compared to SinGan-seg). We need to see if there is significant difference in quality of generated images between the different approaches. This comparison will greatly benefit the readers to understand if the difference in no. of training images Vs. quality of generated images is significantly different.

4.2) Similarly compare the SIFID scores of SinGan-seg with these other approaches.

Reviewer #2: (No Response)

7. PLOS authors have the option to publish the peer review history of their article (what does this mean?). If published, this will include your full peer review and any attached files.

Reviewer #1: No

Reviewer #2: No

---

## [Author Response · Author response to Decision Letter 1]

29 Mar 2022

Response to reviwers letter is attached to this submission.

---

## [Decision Letter · Decision Letter 2]

20 Apr 2022

SinGAN-Seg: Synthetic training data generation for medical image segmentation

PONE-D-21-31343R2

Dear Dr. Thambawita,

We’re pleased to inform you that your manuscript has been judged scientifically suitable for publication and will be formally accepted for publication once it meets all outstanding technical requirements.

Kind regards,

Ruxandra Stoean

Academic Editor

PLOS ONE

Additional Editor Comments (optional):

Reviewers' comments:

Reviewer's Responses to Questions

**Comments to the Author**

1. If the authors have adequately addressed your comments raised in a previous round of review and you feel that this manuscript is now acceptable for publication, you may indicate that here to bypass the “Comments to the Author” section, enter your conflict of interest statement in the “Confidential to Editor” section, and submit your "Accept" recommendation.

Reviewer #1: All comments have been addressed

2. Is the manuscript technically sound, and do the data support the conclusions?

Reviewer #1: Yes

3. Has the statistical analysis been performed appropriately and rigorously? 

Reviewer #1: Yes

4. Have the authors made all data underlying the findings in their manuscript fully available?

Reviewer #1: Yes

5. Is the manuscript presented in an intelligible fashion and written in standard English?

Reviewer #1: Yes

6. Review Comments to the Author

Reviewer #1: The paper has been formatted well. The authors have addressed all my comments and necessary statistical experiments have been performed.

7. PLOS authors have the option to publish the peer review history of their article (what does this mean?). If published, this will include your full peer review and any attached files.

Reviewer #1: No

---

## [Editor Report · Acceptance letter]

22 Apr 2022

PONE-D-21-31343R2 

SinGAN-Seg: Synthetic training data generation for medical image segmentation 

Dear Dr. Thambawita:

I'm pleased to inform you that your manuscript has been deemed suitable for publication in PLOS ONE. Congratulations! Your manuscript is now with our production department. 

Kind regards, 

on behalf of

Dr. Ruxandra Stoean 

Academic Editor

PLOS ONE